METHODS AND RESOURCES

# BehaveAI enables rapid detection and classification of objects and behavior from motion

Jolyon Troscianko[1]*, Thomas A. O'Shea-Wheller[2], James A. M. Galloway[1], Kevin J. Gaston[2]

1 Centre for Ecology & Conservation, University of Exeter, Penryn, United Kingdom, 2 Environment and Sustainability Institute, University of Exeter, Penryn, United Kingdom

* j.troscianko@exeter.ac.uk

## Abstract

Here we introduce BehaveAI, a biologically inspired video analysis framework that integrates static and motion information through a novel color-from-motion encoding strategy. This method translates object movement—direction, speed, and acceleration—into color gradients, enabling both human annotators and pre-trained convolutional neural networks (CNNs) to infer motion patterns while retaining high-resolution spatial detail. Using a range of case studies, we demonstrate how the increased salience of motion information allows for the robust detection of objects that are challenging or impossible to identify reliably from static frames alone, particularly in complex natural scenes. We further demonstrate the reliable classification of different behaviors in animals and single-celled organisms. Additionally, the framework supports flexible hierarchical model structures that can separate the tasks of detection and classification for optimal efficiency, and provide individual tracking data that specifies *what* is present *where* and what it is *doing* in each frame. The framework makes use of the latest deep learning architecture (YOLO11), combined with a semi-supervised annotation workflow. Together with salient motion information, these features can dramatically reduce the effort required for dataset annotation such that reliable models can often be made within an hour. Moreover, smaller annotation datasets mean that model training can be achieved on conventional computers without dedicated hardware, thereby improving accessibility. The motion encoding approach is also computationally lightweight, and can run in real-time on low-end edge devices such as a Raspberry Pi. We release the framework as a free, open source, and user-friendly package.

## Introduction

Videos convey rich information about *what* is present *where*, and crucially, what it is *doing*. This capacity to record spatiotemporal relationships and behavior makes the

**Data availability statement:** GitHub repository for latest software download and user guide: https://github.com/troscianko/BehaveAI. Software release snapshot v1.2.0 at point of publication: https://doi.org/10.5281/zeno-do.18232555. Case study data (figshare): https://doi.org/10.6084/m9.figshare.30531116. Sperm and pigeon videos are freely available through other repositories (cited in the manuscript).

**Funding:** JT, JG, and KJG were funded by the Natural Environment Research Council (UK) grants NE/W006359/1 and NE/Z000114/1. The funder had no role in study design, data collection and analysis, decision to publish, or preparation of the manuscript. URL: https://www.ukri.org/councils/nerc/.

**Competing interests:** The authors have declared that no competing interests exist.

**Abbreviations:** CNNs, convolutional neural networks; fps, frames-per-second.

use of video ubiquitous in society and indispensable for scientific research observing animal behavior and ecological interactions. More widely, videos are used across sectors such as animal husbandry and farming, industrial quality control, security and surveillance, and medical diagnostics [1]. Nonetheless, despite decades of advancement in video processing, efficient and accurate quantification of complex motion information—critical for object detection and classification—remains a significant computational challenge, particularly in unconstrained visual scenes.

Convolutional neural networks (CNNs) have revolutionized the analysis of static visual information, with modern image classification, object detection, and segmentation tools demonstrating remarkable accuracy [2,3], and gaining widespread use in the life sciences [4,5], and wider society. Crucially, they can be trained effectively even with moderately-sized datasets through techniques such as transfer learning and data augmentation. These tools excel at recognizing objects based on their spatial features: shape, texture, color, and pattern, treating video analysis as a sequence of independent image processing tasks [4]. However, while this approach has yielded substantial progress, it inherently neglects motion, which is critically important in many visual detection and classification tasks. For example, camouflaged animals can easily evade detection by blending in with their static surroundings, but they can become trivially easy to detect as soon as they move [6]. Patterns of biological motion are also critical for recognizing and classifying behavior, as illustrated by the relative movements of a few dots that are instantly recognizable as a human walking [7].

Tools that integrate motion information for detection and classification remain less widespread than static variants, and are generally more limited in scope. DeepEthogram can effectively determine patterns of animal behavior through static and motion information [8]; however, the tool is not suited to the detection, classification, and tracking of animals in typical natural scenes. LabGym detects temporal changes in animal outline shape to predict its behavior, and supports tracking the behavior of multiple individuals [9], yet the model training process is complex, particularly when detecting objects against natural backgrounds. Pose estimation tools such as DeepLabCut, SLEAP, and DeepPoseKit are effective for determining the spatio-temporal movements of whole animals and their limbs [10–12], but these data don't immediately tell us what the animal is doing without considerable further interpretation [13]. Consequently, downstream pipelines such as Keypoint-MoSeq are required to convert pose data into behavioral classification [14]. These additional training workflows are typically time consuming and do not scale well with large numbers of animals, limiting their accessibility.

Critically, tools that rely on outline shape or pose estimation fail when bodies and limbs cannot easily be resolved, e.g., due to small limbs and video resolution limits, motion blur, fast movement, complex backgrounds, occlusion, or limb transposition (the limbs of nearby individuals causing confusion). These are common circumstances in real-world videos, particularly considering that most animals have evolved coloration, movement, and background selection in order to evade detection. Alternative 3D CNNs such as Inflated 3D ConvNet can combine static and optic flow

streams for effective video classification of human actions [15,16], although these systems classify the content of an entire video frame, and are not readily suited to object classification and tracking tasks. As such, they have not received widespread use in the behavioral and ecological sciences when compared to static CNN frameworks despite the fields' heavy reliance on video data [13] and common need to quantify behavior.

Drawing together visual information from both movement and static appearance has the potential to transform behavioral quantification, yet this remains computationally challenging. The mammalian visual system achieves this by using segregated neural pathways for different visual tasks. The ventral stream primarily processes form, color, and object identity—the "what" of the scene. The dorsal stream deals with motion, spatial relationships, and guiding action—the "where" and "how" ([17], Chapter 1.3.3). These functionally distinct processing streams are then integrated to create a complete percept. Inspired by this segregated but integrated processing, we present a novel framework specifically designed for robust object detection, behavioral classification, and individual tracking from video data. Our framework converts motion information into false colors that allow both the human annotator and CNN easily to identify patterns of movement (Fig 1). In contrast to motion history images [18], outline shape changes [9] or optical flow based kinematics [15,18], our color-from-motion strategy provides information on the direction, speed, and acceleration of movement through color gradients, with different colors reaching further back in time. This motion encoding method uses a trivial amount of computational power and shifts the motion interpretation to the pre-trained CNN while also retaining high-resolution spatial information, meaning that the framework can achieve real-time video processing on low-end "edge" devices. The system can also integrate conventional static (color, pattern, form) information together with hierarchical processing, meaning that the tasks of detection and classification can use motion and/or static visual information, much like the mammalian visual system. We further introduce a semi-supervised annotation workflow for rapid and efficient "auto-annotation".

Here, we describe the framework and demonstrate its ease-of-use and efficacy in a wide range of challenging object detection and classification tasks. Although we provide some comparisons, we have not focused on directly equivalent benchmarks between our methods and others due to the dangers of attempting to over-generalize AI model utility [19]. Indeed, the CNN architecture that our framework relies on has been demonstrated elsewhere [20]. Instead, our case studies focus on BehaveAI's potential to transform the capabilities of AI models using video datasets likely to represent real-world biological challenges (with natural, complex backgrounds), and show how effective classification and tracking can be achieved quickly and reliably.

## Results

Our case studies test the performance of the BehaveAI framework in a range of challenging tasks, and we use these to outline key advances. First, movement is typically far more salient than static visual cues, increasing the signal-to-noise ratio of objects against their backgrounds, and making the models more independent of background appearance or lighting conditions (Figs 1–3). The grass moth case study shows how a motion-based model can significantly outperform static-based models for equal annotation effort, with precision and recall of 0.966 and 0.969 (motion, 98.3% correct validation inferences, i.e., 115 correct detections for 2 false negatives) versus 0.792 and 0.544 (static, 67.5% correct validation inferences). We give the precision and recall metrics as provided in the final epoch of model training by YOLO. Precision indicates how many detections were correct, and high numbers imply few false-positives. Recall reflects the model's ability to identify all instances of objects, and high numbers imply that few objects are missed. The pigeon case study similarly achieves comparable performance to Chan and colleagues [4] who used 1,587 training images, while we used just 295, despite us adding an additional behavior class (see Fig 4). The sperm motility case study can classify subtle differences in sperm movement with precision 0.793, and recall 0.801 based on 333 annotated motion frames, while using static frames alone reduced model performance to 0.610 and 0.594, respectively. Other studies show that classifying sperm defects from static frames is considerably more difficult. For example, Thambawita and colleagues [21] use annotations from over 29,000 frames to achieve precision 0.571 and recall 0.228 with the same dataset.

 

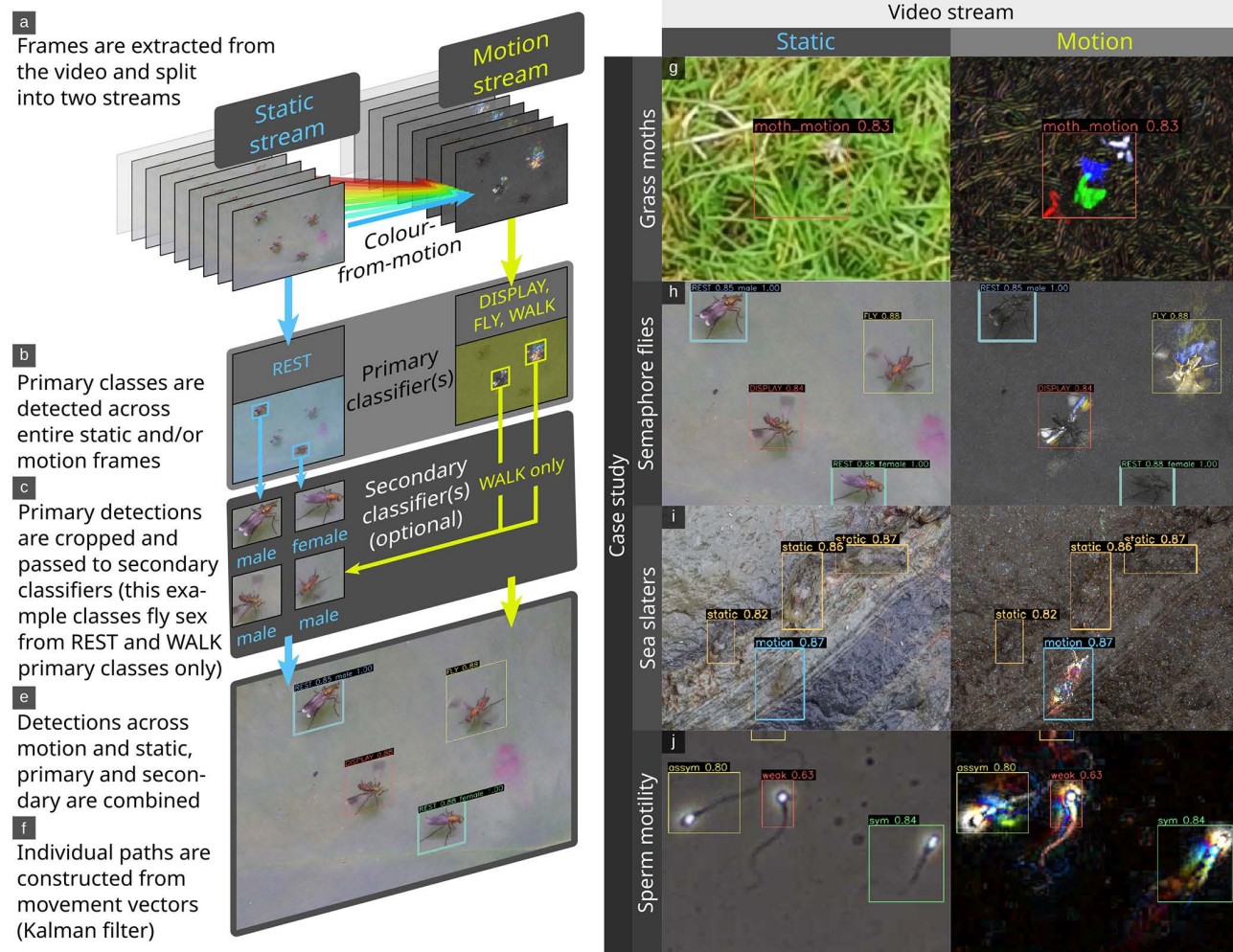

**Fig 1. Illustration of the framework's structure.** Users can specify primary classes and secondary classes across both static and motion video streams **(a–f)**. The right panel shows examples of static and color-from-motion frames from case studies. The color-from-motion trails create characteristic patterns that reveal movement and behavior. The grass moth (Crambidae) is nearly undetectable from still frames due to its color, and motion blur, but it is highly salient in the motion stream **(g)**. The semaphore fly (*Poecilobothrus nobilitatus*) example shows how motion information can easily disentangle behaviors that are often identical from static frames (e.g., "fly" and "display" appear identical in the static frame, but different in motion). This example also showcases hierarchical classification, with secondary classifiers determining the sex of the flies **(h)**. Sea slaters (*Ligia oceanica*) are highly camouflaged when static, and salient when moving, resulting in motion models that make far fewer errors (but cannot detect stationary individuals) **(i)**. Human sperm have been classified based on their swimming movement with either symmetric (typically resulting in fast, straight movement), asymmetric (typically resulting in slow, circling, exploratory movement), or weak (twitching, vibrating etc…) strategies. These swimming strategies can be determined without tracking individuals, which is difficult in complex, debris-filled videos **(j).**

A second major benefit of the framework relates to the fact that most animal (and single-cell/subcellular) behavior is characterized by biological motion, and by training the model directly on the motion cues—not just speed and direction, but also spatial patterns of acceleration and deceleration—we can bypass the complex and computationally demanding steps required to predict behavior from pose estimation or whole-individual-tracking. This is highlighted by the sperm motility example (above), where tracking the progress of individual sperm (which is difficult when multiple individuals crowd together) is not required in order to assess the frequencies of different sperm motility types in a given sample (Figs 1, 4, and 5).

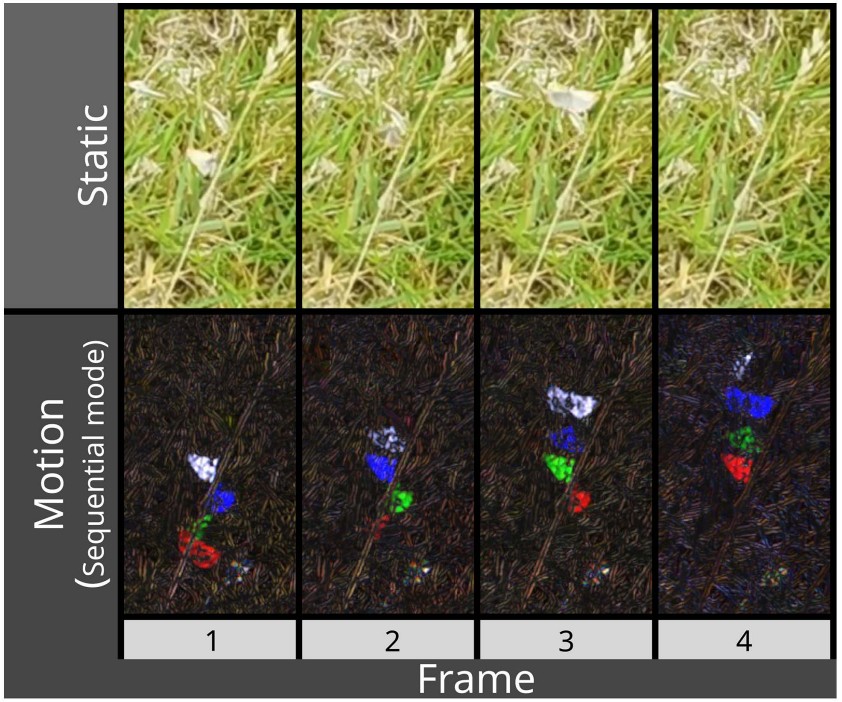

**Fig 2. A grass moth (Crambinae) in flight, cropped over consecutive frames.** The moth is barely visible in static frames (top), but becomes highly visible in motion (using sequential mode in this case).

Third, using the BehaveAI framework, behavior can be classified from statically-identical frames, as highlighted by the pigeon example, where a motionless pigeon often has an identical pose to a preening or walking individual (Figs 1 and 4). Similarly, in the semaphore fly example, a flying insect can appear identical to a courtship displaying insect, and a walking insect is visually identical to a resting insect in a static frame (Fig 1h). In this latter case, incorporation of information allows behaviors to be classified with precision 0.895 and recall 0.894 (correct inferences for display: 97%, fly: 94%, and walk: 94%).

Fourth, with the BehaveAI framework, movement can often be detected and classified from a far lower spatial resolution than static classifiers. The grass moth scaling test (Fig 3) shows how the motion-based model can perform very well: precision and recall of 0.945 and 0.949, respectively, (>96% correct validation inferences) when the static targets' bounding boxes are just 4.3 pixels across on average, and this drops to just 0.813 and 0.795 (82% correct validation inferences) when targets are only 2.0 pixels across on average. Static model performance at these scales is 0.633 and 0.538 (36.4% correct) and 0.225 and 0.3117 (27.3% correct), respectively.

Finally, the framework's color-from-motion strategy is extremely computationally efficient, enabling real-time detection and classification on low-end devices. We demonstrate the framework running in real-time on a basic Raspberry Pi 5 at 9–11 frames-per-second (fps) with inference being performed on input video resolution of 640×480 pixels. A standard laptop computer without graphics card acceleration achieved inference speeds of 30–31 fps on 1,280×720 pixel live video streams.

## Discussion

The BehaveAI framework combines a novel color-from-motion strategy with state-of-the-art deep learning architecture in an integrated pipeline. Our results showcase the accuracy, flexibility, ease-of-use, and computational efficiency of the framework for detecting and classifying objects from their movement. Specifically, we show how the framework can

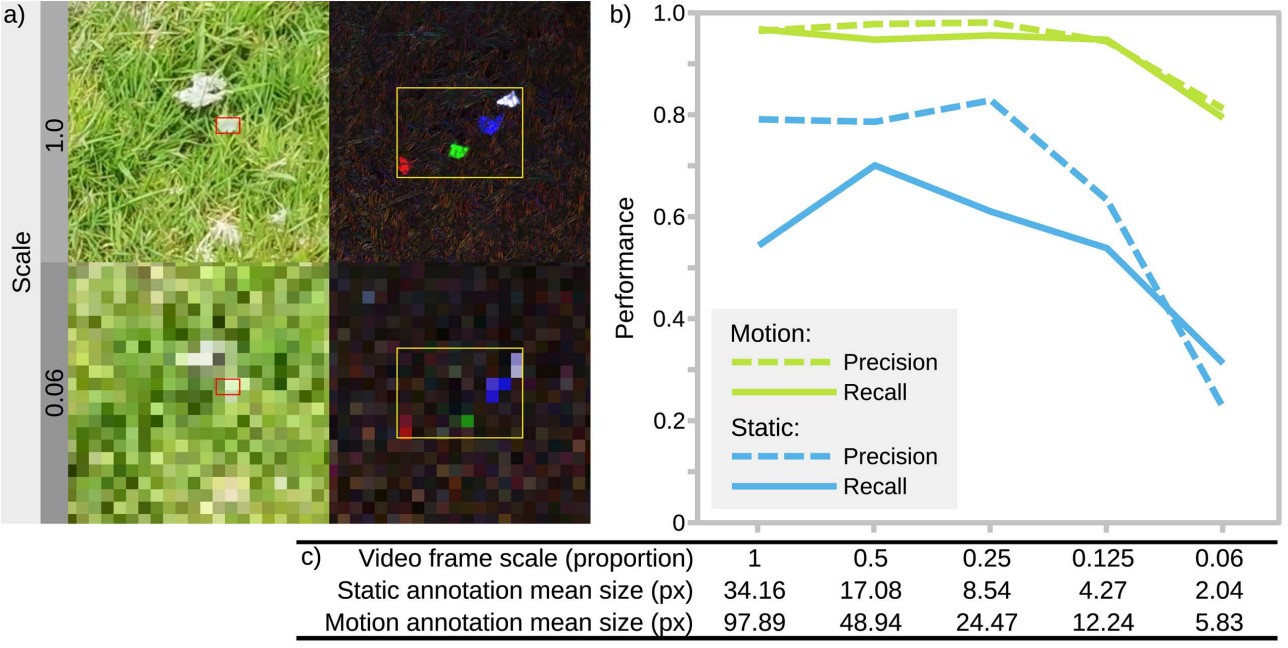

c)

| Video frame scale (proportion) | 1 | 0.5 | 0.25 | 0.125 | 0.06 |
|---|---|---|---|---|---|
| Static annotation mean size (px) | 34.16 | 17.08 | 8.54 | 4.27 | 2.04 |
| Motion annotation mean size (px) | 97.89 | 48.94 | 24.47 | 12.24 | 5.83 |

**Fig 3. Motion and static classifier performance across different target sizes.** Grass moth annotation images were progressively scaled down, and models re-trained on each resolution. **(a)** shows cropped regions with unscaled (1.0) and 1/16th scaled examples, with boxes showing the bounds of the static (red) and motion (yellow) grass moths. Movement creates a substantially larger and more salient target for detection. **(b)** shows model metrics at different scales; the motion model performs consistently high at all scales, but the static model performance decreases markedly with scale. The performance of the static model is over-estimated here because moths are simply not visible in a substantial number of frames. **(c)** lists the scales and respective mean annotation box sizes (in pixels). Videos are 1,920 × 1,080 pixels at scale = 1.0. The data underlying this Figure are available here: https://doi.org/10.6084/m9.figshare.30531116.

achieve comparable or greater inference performance from far smaller annotation training sets than alternative methods, and is able to track objects that might not be possible with other methods. This, in turn, brings further benefits: reducing training time, allowing models to be trained and deployed on low-end computers, simplifying analysis, and reducing energy consumption. This also eliminates the need to purchase cloud-based compute services, reducing costs and barriers to entry. Notably, the framework also excels at tracking small, camouflaged moving objects—just two pixels across—that cannot be reliably detected from still frames. This means that a motion-based strategy could effectively cover a much larger spatial area and still detect targets with far greater accuracy than a static model. Alternatively, it shows how video streams can be reduced in size substantially without affecting performance, thus increasing computational efficiency.

Determining *who* is doing *what*, *where*, and *when* will often require a more complex set of classification tasks than can easily be achieved in a single classifier. The BehaveAI framework's biologically inspired separation of motion and static video streams, together with hierarchical processing, allows for considerable flexibility in structuring the pipeline to meet individual needs. This is highlighted in the semaphore fly example, where flies are first detected by primary models that search across whole frames for specific motion (walking, flying, displaying), or static appearances (rest). Once detected by a primary model, the flies are passed to a secondary model that determines the sex of the fly from the cropped region, increasing computational efficiency and reducing redundancy. This maximizes the efficacy of the initial detection task—whether flying quickly or completely still—and only then determines the sex based on subtle visual cues.

Manual annotation of objects in images/videos is a crucial bottleneck in the deployment of deep learning models, typically requiring considerable time investment before performance can be assessed. The sheer diversity of systems, organisms, and behaviors observed within the biological sciences means that generalized "foundation" models often fail

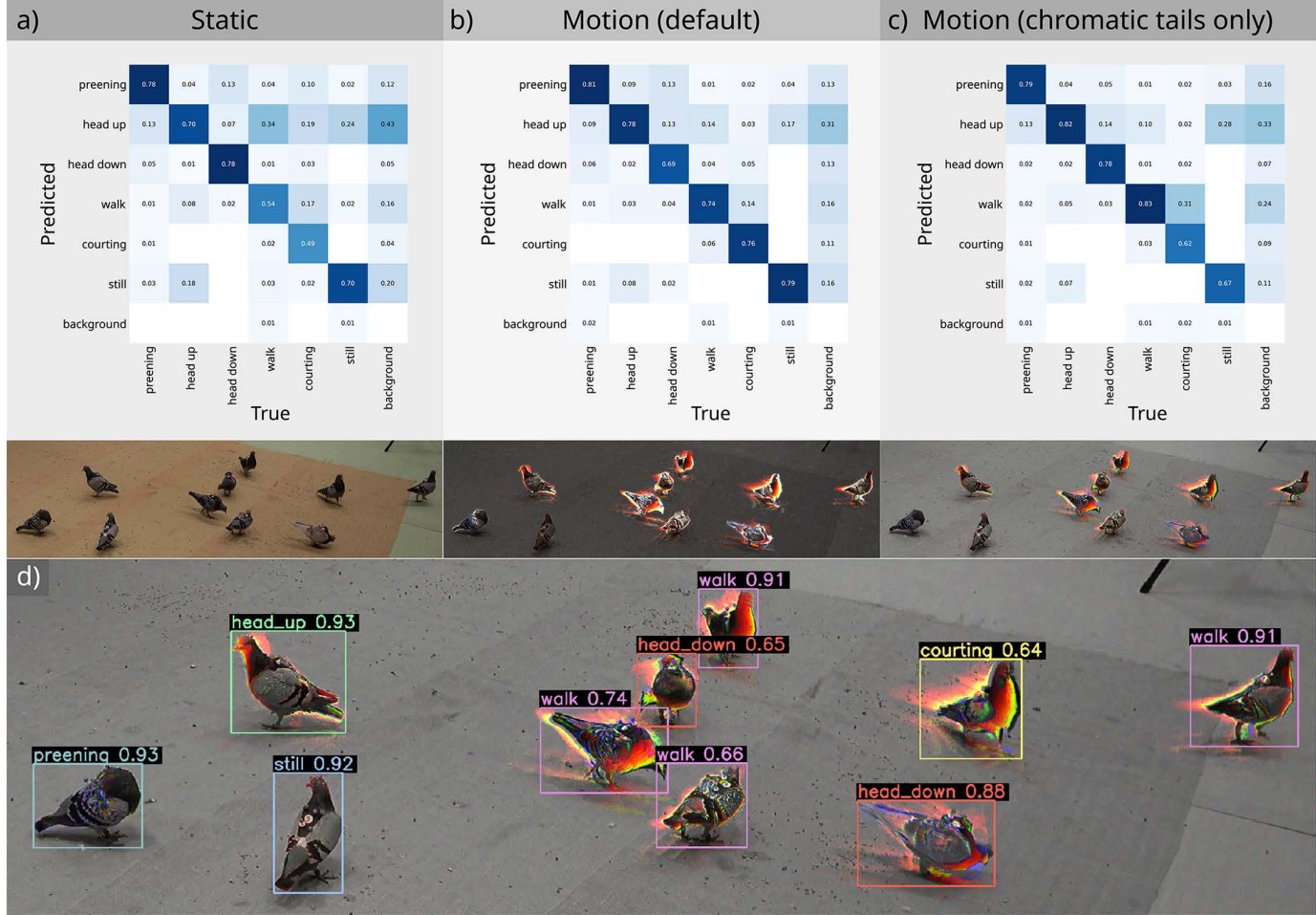

**Fig 4. Direct comparison of pigeon behavioral classification using different motion strategies. (a)** shows results for conventional 'static' models, while **(b)** and **(c)** show models based on alternate colour-from-motion strategies. **(d)** shows an auto-annotated frame using an exponential strategy with chromatic tails only, correctly classifying all of the pigeons and their behaviors. Numbers show the auto-annotation confidence. The data underlying this Figures are available here: https://doi.org/10.6084/m9.figshare.30531116.

to transfer between systems, requiring researchers to develop bespoke models for their specific use cases [22]. Efforts have therefore focused on optimizing this process, rather than attempting to build generalized datasets [19,23]. The BehaveAI framework uses a semi-automated annotation workflow, which—combined with the above increase in salience from motion cues—results in a far more efficient annotation process. By utilizing model predictions to augment annotation, the number of manually drawn annotations required is considerably reduced, while cases where the model performs poorly are highlighted, enabling the user to target additional data for inclusion only as needed. As above, this reduces the volume of data required to develop a robust model and limits training and deployment costs downstream. The grass moth example (Figs 1 and 2) details how—within one hour—roughly 550 frames can be annotated and a model trained that achieves remarkable tracking performance. This is despite the videos having substantial background motion due to grass swaying in the wind and the camera moving. We also include tools for inspecting the annotation dataset (in order to fix errors or re-classify annotations), and for rebuilding the motion annotations in order to compare the efficacy of different color-from-motion strategies. Together, this makes for a user-friendly workflow that maximizes efficiency and accessibility.

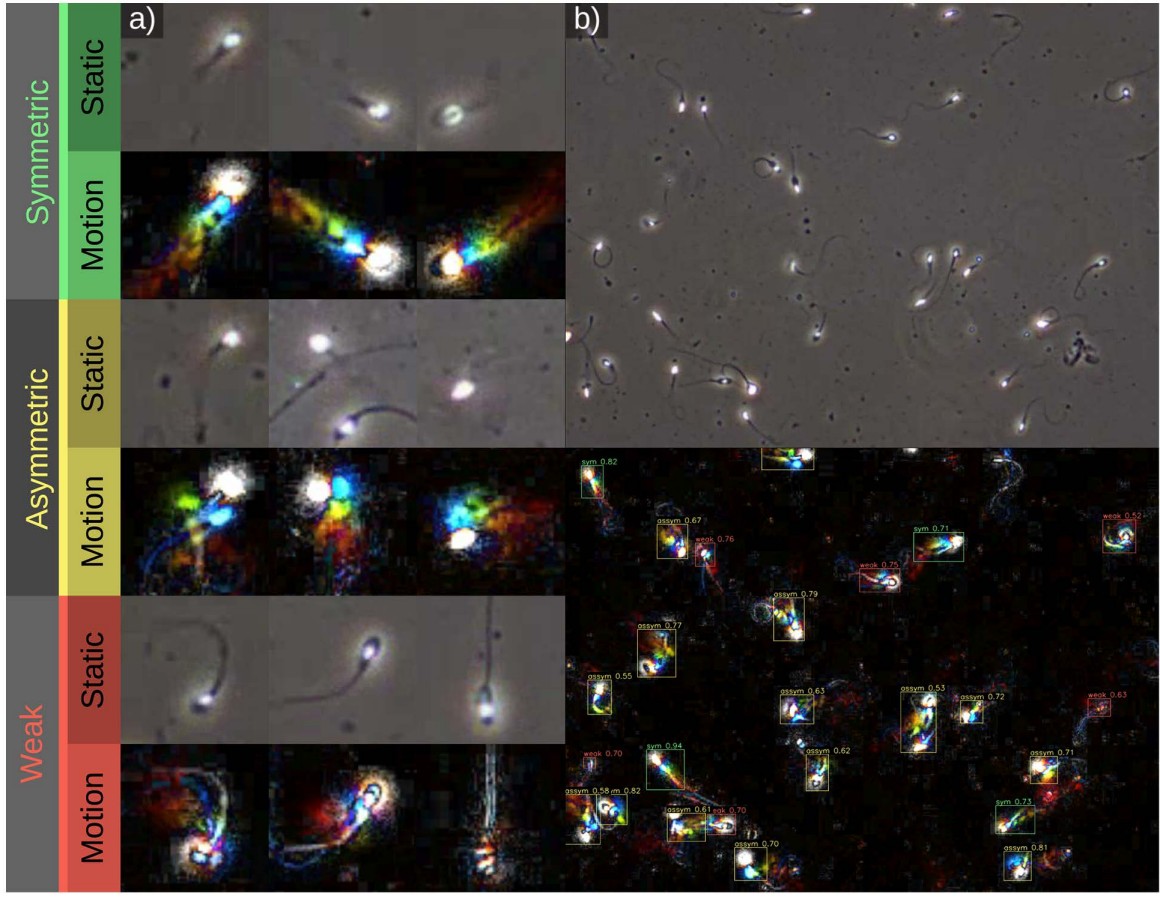

**Fig 5. Examples of static and motion streams for the three types of sperm motion being classified. (a)** shows close-up examples of the different sperm movement behaviours being classified in static and colour-from-motion views. Symmetric movement results in a linear, consistent motion trail behind the head. Asymmetric motion results in a zig-zag, irregular motion trail, while weak motion (vibrating/twitching sperm) creates motion energy without a clear trail. **(b)** shows a full-frame example from the dataset with classifications and confidence shown in the bottom panel.

Implementing edge-deployable BehaveAI models on low-end hardware opens up a range of opportunities for real-time monitoring and interaction with behavior in the field or lab. We demonstrate that the framework's inference processes can run on Raspberry Pi 5 units, with YOLO11 models being converted to NCNN format for computational efficiency. The enhanced salience of motion information means that video frames can be reduced in size substantially, thus increasing speed without impacting performance. This demonstration highlights the versatility of the BehaveAI framework for deployment in embedded and remote scenarios.

The current BehaveAI framework provides unique opportunities for efficient and robust behavioral quantification, yet there are a number of avenues to explore for future enhancement. Integrating existing annotated datasets is not currently straightforward because of the need to create motion frames from the preceding video sequence, in addition to the boundaries for motion annotations differing from the static; ongoing development will aim retroactively to extract motion information from such data, although this would only be possible for datasets where full videos with annotations are available. Beyond this, we aim to incorporate support for segmentation, pose estimation, and oriented bounding-box annotations. Together with these enhancements, the open source code and documentation provided here will enable the incorporation

of our motion strategy into a wide range of aligned deep learning frameworks. The utility of the color-from-motion strategy—or similar solutions that encode image difference over a range of different timescales—should also be explored in fields beyond the biosciences. For example, its ability to quantify acceleration and deceleration of objects within a single frame could be valuable in machine vision tasks such as robotic applications or autonomous vehicle control.

We aim to make the BehaveAI framework as accessible as possible, and release the tools as a free and open source software package. We also include a range of tutorial videos and sample data. The framework's flexible structure means that switching the pipeline to use alternative deep learning architectures is straightforward, allowing the system to be upgraded easily to keep pace with the latest advances in deep learning.

## Case Studies

### Grass moths

Our grass moth dataset presents a challenging but common detection task in ecology: tracking small animals that are motion-blurred and often occupy just a few pixels, traveling fast against complex, moving backgrounds with an abundance of wind-blown vegetation distractors of a similar size, color, and shape. In any given static frame, the moths are barely visible, but their motion is typically highly salient (Figs 1 and 2). The dataset is made up of 32 short video clips of individual grass moths (Crambidae, unknown species) taking off as they are approached on foot in a grazed pasture field in the UK with varied lighting conditions (sunny to cloud-covered). These were filmed using a hand-held smartphone (Huawei p30, 1,920 × 1,080, 60 fps). We used the "sequential" motion strategy with a YOLO11n model, 50 epochs per training round (full settings and data are available in the figshare repository [https://doi.org/10.6084/m9.figshare.30531116]).

First, we use the dataset to demonstrate the speed with which an effective moth tracking model can be built. Training initially used 77 annotated frames from 6 videos (38 additional annotation frames were automatically set aside for validation), after which the first model was trained. Following this, semi-supervised annotation ("auto-annotation") was used to add another 168 training annotation frames from 7 videos (plus 52 set aside for validation), and a second model was trained. At this point, auto-annotation was making very few errors, and the final round of annotation focused on correcting and adding blank frames that had been auto-annotated with false positives, amending false negatives, and adding low-confidence auto-annotated hits, yielding another 194 training annotation frames (plus 43 set aside for validation) from 6 new videos. Only a small fraction of these annotations were manually drawn. This entire annotation process (548 annotated frames, of which 57 were blank frames, including two rounds of model training), took 61 min to complete. The confusion matrix of the final model shows that 114 moths were correctly detected in the validation set (98.3%), with 2 (1.7%) false negatives, and 7 false positives; precision: 0.966; recall: 0.977. Note that these false positives would be easy to exclude post-processing by filtering out any short tracking paths based on tracking ID.

In order directly to compare the performance of motion-based classification to a conventional static classifier, we annotated the same grass moth videos from the static stream. Given the moths' speed, the motion annotations must be drawn over a considerably larger area than the static ones, so separate annotations were made. The moths are often not distinguishable from their background in static frames (e.g., wings folded up or down, with motion blur and similar color and shape to background features, see Fig 2), and were only included if they were visible (meaning the static model could never match the motion model's ability to detect the moths in all frames). The motion dataset used 415 frames for training and 133 for validation, from 19 videos; the static dataset used 460 frames for training and 103 for validation. YOLO11s models with 150 epochs were used for training. The motion model achieved precision and recall of 0.966 and 0.969, respectively (S1 Fig; of 115 correct classifications, there were 2 false negatives and 14 false positives). The static model achieved precision and recall of 0.792 and 0.544, respectively (S2 Fig; of 52 correct classifications there were 25 false negatives and 32 false positives). But as noted above, this over-estimates practical performance of the static model as the moth was not distinguishable from its surrounds in a large proportion of frames, making continuous tracking from static frames far more problematic than from motion.

Next, we tested how target size affects model performance. Images in the above training sets were reduced in resolution to 1/2, 1/4, 1/8, and 1/16 (0.5, 0.25, 0.12, and 0.06) of their original scale, and models were re-trained using the same settings. Rescaling was performed using the batch conversion tool in ImageJ v1.52, with bicubic interpolation. The results are shown in Fig 3. The motion model's precision and recall were greater than 0.945 and 0.949, respectively, down to 1/8th scale (image dimensions 240 × 135 px, target dimensions 4.27 px), and reduced to 0.813 and 0.795 at 1/16th scale (image 115 × 64 px, target 2.04 px). The static classifier performed worse, at all scales, with precision and recall greater than 0.633 and 0.538 down to 1/8th scale, and 0.225 and 0.312 at 1/16th scale.

**Pigeon behavior—3D-POP dataset.** We used a subset of the publicly available pigeon behavior dataset [3D-POP, [24]] for comparing performance with and without motion strategies. We annotated 295 training images, and 70 validation images, taken from 4 training videos (considerably fewer than Chan and colleagues [4], who used 1,587 training images). We specified behavior classes similar to those of the original publication and those used for benchmarking by Chan and colleagues [4]. However, we used the broader definition of "courtship" to refer to all male courtship behavior, rather than the "bow" alone now that we are not constrained to cross-verify with 3D body-tracking data. We also added a "still" class for motionless birds. Given that the birds were slow-moving against a uniform background, annotation boxes drawn for motion were also suitable for static frames, allowing for direct comparisons without redrawing annotations. Three classifiers were trained from the same annotation data by altering the image processing strategy: "static" (conventional motionless frames, Fig 4a), "motion (default)" (exponential motion strategy, Fig 4b), and "motion (chromatic tails only)" (exponential strategy with chromatic tails only set to true, Fig 4c). Each was trained with 100 epochs using YOLO11s. Confusion matrices for each strategy are shown in Fig 4. The static model achieved precision and recall of 0.651 and 0.677, the default motion model achieved 0.788 and 0.780, and the motion-tails-only model achieved 0.746 and 0.787, respectively. Chan and colleagues [4] report a precision and recall of 0.77 and 0.70, respectively, with this dataset, although note that the behaviors are slightly differently structured.

## Sperm motility classification

Quantifying sperm motility is important for assessing human fertility, and a range of tools have been developed to automate the process. Nevertheless, the task remains difficult, particularly given the wide range of imaging systems, tail movements that are often difficult to track, and samples with a considerable degree of debris and visual noise. We used the VISEM video dataset of human sperm [21,25] for training a motion classifier to distinguish between three characteristic sperm motion strategies; symmetric, asymmetric, and weak motion (see Fig 5 for examples). The training set consisted of annotations from 333 frames using 16 videos, plus 74 validation frames. A YOLO11m model was trained with 200 epochs, resulting in a precision of 0.793, and recall of 0.801 (S3 Fig). A key advance here is that the motion behavior of a sample can be reliably ascertained from a single frame without needing to track the movement of specific individuals (which is problematic in complex scenes).

Training the model using static frames alone (and all other parameters kept the same) yielded precision and recall of 0.610 and 0.594, respectively (S4 Fig).

## Sea slater detection

Sea slaters (*Ligia oceanica*) are littoral-zone isopods that exhibit polyphenic coloration (i.e., individuals vary markedly in their patterns and colors), they use adaptive color change to match their backgrounds, microhabitat selection based on habitat color and shape, and fast, intermittent motion [26]. Their extremely effective camouflage therefore presents challenges for visual detection, particularly as the human annotator will often fail to detect them until they are revealed by their movement. We used a video dataset of slaters filmed against their natural backgrounds around the coast in Falmouth (UK), and specified a primary static class ("static"), and a primary motion class ("moving"). Training used annotations from 130 frames, with 38 for validation from 12 videos. Annotations were only included if the slater was clearly visible to the

annotator, and this decision was applied independently to the visual and static streams; slaters that were small, perfectly matched to their surrounds, or out of focus were typically easier to detect in the motion stream, and were not included in the static. We avoided repeatedly entering the same individual slater (and over-fitting the model for specific individuals) by using gray boxes to conceal them in subsequent frames unless they moved (see below). YOLO11s motion and static models were trained for 200 epochs. The static model achieved precision 0.875 and recall 0.846 (S5 Fig). For 125 correct classifications, there were 23 false positives and 11 false negatives. The motion model had precision 0.834 and recall 0.922 (S6 Fig). For 96 correct classifications, there were 15 false negatives and 7 false positives.

The case studies here all use a default allocation of annotated frames to either training or validation datasets, i.e., each annotated frame was given a 0.2 probability of being sent to the validation set. As such, each validation frame is likely to be from a video that was also included in the training dataset (with shared object and scene appearance), making them non-independent. We therefore ran a second test with validation and training annotations taken from mutually exclusive videos. Model outcomes were very similar with precision and recall for static of 0.908 and 0.859 (S7 Fig); and motion of 0.893 and 0.876 (S8 Fig).

## Semaphore flies

Semaphore flies (*Poecilobothrus nobilitatus*) have complex courtship behaviors that present a challenge for tracking and classification, with fast-moving limbs (wings and legs are often a blur at typical filming speeds), and fast flight. This example highlights some of the more advanced options available in the BehaveAI framework when working with more complex scenarios. The video database consisted of wild flies, filmed at 60 fps using a Sony a6400 with 60 mm prime macro lens. Moving flies were highly salient in the motion stream, and three primary motion classes were specified (walk, fly, and display). When the flies were not moving they were difficult to spot in the motion stream, so we also added a primary static class (rest) to find them from the static stream. However, the static classifier would be able to see all the cases of flies walking, flying, or displaying that are not classed as rest. This would likely confuse the static classifier because a walking fly looks a lot like a resting fly. So we added motion_blocks_static = true to hide all the instances of walking, flying, or displaying flies from the static classifier.

We also wanted to determine the sex of each fly from its wing markings, so added male and female as secondary static classes. However, when displaying or in flight these wing markings were not visible, so we could tell the model to ignore running the secondary classifier for these cases (ignore_secondary = display, fly). Finally, flies will often be detected by both the motion and static classifier, but the motion classifier will be more reliable and make very few false positive errors, so we set this to be the dominant stream for detections (dominant_source = motion). This only affects the video output— data from both streams are saved in the output.

Annotations from 524 frames were used for training, and 129 for validation, from 19 videos. YOLO11s classifiers were trained for 150 epochs (YOLO11s-cls for secondary models). The primary motion classifier achieved precision and recall of 0.933 and 0.932, respectively (S9a Fig), while the static model achieved 0.985 and 0.997, respectively (S9b Fig). Secondary models (determining sex from the static images) achieved a reported accuracy from YOLO of 1 (i.e., correctly categorized all 57 males and 58 females while at rest S9c and S9d Fig).

Sample videos also highlight the individual tracking efficacy, with the Kalman filter-based tracker successfully tracking the flies for extended periods despite instances of individual paths crossing.

## Edge performance

We tested real-time ("edge") performance of BehaveAI on a low-end device (Raspbbery Pi 5 running Debian 13 Trixie Pi OS 64-bit, with wide-angle OV5647 camera). We trained a primary motion classifier to detect the behavior of adult *Tenebrio molitor* beetles on an oatmeal substrate, as either move, bury, or interact ("beetle"' example). We also created an artificial hierarchical demonstration ("camo_targets") from sheets of card printed with different backgrounds with square or

triangular paper targets printed with samples of background. The targets could be moved with magnets from underneath while being videoed from above. When stationary, the targets are effectively undetectable (being a perfect sub-sample of their complex surrounds), but the model could identify both shape and movement strategy from their motion cues. We trained a model with two primary motion classes ("move" and "turn"), and two secondary motion classes ("triangle" and "square"). The Raspberry Pi processed both beetle and camo_targets models at between 9 and 11 frames-per-second with 640×480 resolution video and NCNN enabled. The same models ran at between 30 and 31 fps with a higher resolution of 1,270×720 pixels on a conventional laptop with USB web camera (Framework Laptop 13 with AMD Ryzen 7040 Series CPU, Ubuntu 24.04) without GPU acceleration. The data for these projects are available in the figshare database.

## Methods

### Color-from-motion

The framework operates by splitting video input into two parallel processing streams: the "static" stream treats each frame as a conventional still color image that provides shape, color, and textural information. The "motion" stream presents temporal information using false colors that can intuitively be interpreted by both humans and CNNs to see patterns of movement (including speed, acceleration, and direction) over short timescales.

There is a range of user-selected options for adjusting the color from motion strategy to suit various use cases. Two different motion strategies are available; the "exponential" method calculates the absolute difference between the current frame and the previous frames, exponentially smoothing over successive frames to show different temporal ranges in different color channels. With this mode, a moving object creates a white "difference" image that leaves behind a motion blur that fades from white to blue, to green, to red. Increasing the exponential smoothing weights allows this method to show events further back in time at no extra computational processing or memory costs when running the classifier because there is no need to re-load any previous frames. This mode is better able to convey changes in speed within each frame; accelerating objects will outpace their red tail, creating a blue-to-green streak, while deceleration will allow the red tail to catch up, creating yellow-to-red tails. A further option ("chromatic tails only") removes the initial white difference information, preserving more static luminance information. This mode preserves the greatest proportion of luminance information, so is likely to suit tasks where detailed texture or form is key.

The "sequential" method uses discrete frames rather than exponential smoothing, with colors coding the differences between the previous three frames (white, blue, green and red going back through time respectively), and is suited to classifying movements over this short range of frames and preserving more spatial information from previous frames (e.g., rather than a smooth tail, a flying animal's characteristic wing shapes will remain visible over all four frames). The motion false color is then optionally blended with the luminance channel to provide greater context—combining motion information with static luminance. The exponential weightings, false-color composition, and degree of luminance blending are all user-adjustable. Frame skipping can also be used to perform measurements over a larger number of frames (longer timespan) at no additional processing costs (e.g., suited to slow-moving objects whose behavior is more apparent from faster video playback). Settings can be altered following annotation, and the motion annotation dataset rebuilt using the new settings. The framework automatically detects changes in these settings if a model has already been trained, and asks whether the user would like to re-build and re-train with the new settings (Fig 6).

### Classifier structure

We use the latest generation of YOLO classifiers [20] for rapid and effective detection and classification from static and/or motion video streams. The user can select a model version (e.g., YOLO 11 or 8), and model size; larger models can improve accuracy, but require more computational resources. The nano ('n') or small ('s') models are generally sufficient. The framework supports both parallel and hierarchical processing of the static and motion streams. "Primary" classifiers

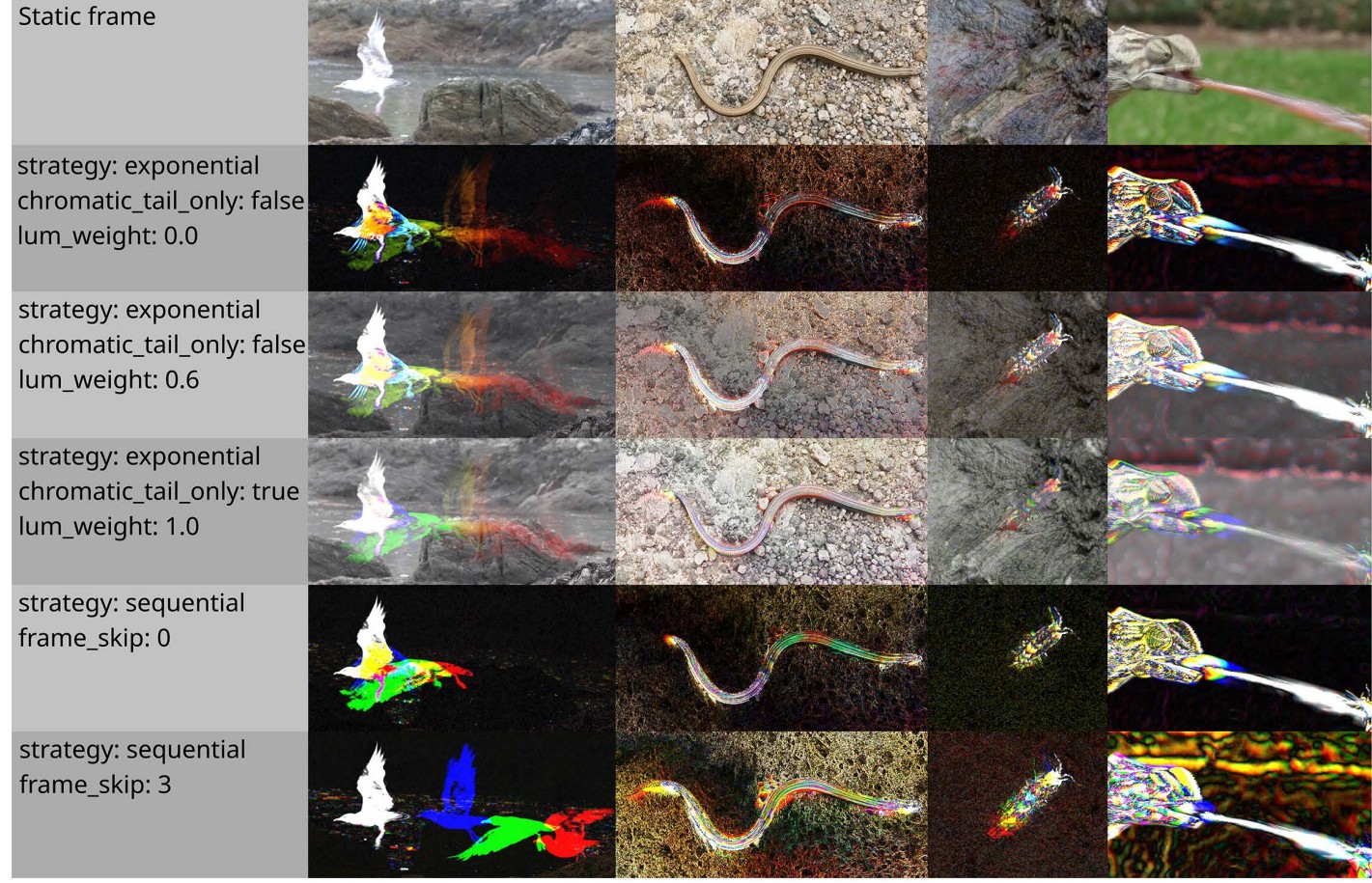

**Fig 6. A selection of color-from-motion strategies and parameters available in the BehaveAI framework.** Examples from left to right are a herring gull (*Larus argentatus*) taking flight, a common slow-worm (*Anguis fragilis*), a camouflaged sea slater (*Ligia oceanica*) moving, and a flat necked chameleon (*Chamaeleo dilepis*) hunting an insect. The motion-coding parameters can be altered after building the annotation training set, and new models trained, allowing users to test which parameters best suit their system without redoing annotations.

search each entire static and/or motion frame for objects (including objects of multiple types or "classes"); objects detected by a primary classifier are then automatically cropped and sent to "secondary" static and/or motion classifiers (if a hierarchical model is specified). This allows the tasks of detection and classification to be performed using whichever stream (motion or static) is most appropriate. Users can specify the model structure with primary and (optional) secondary classes, together with the parameters for annotation, motion processing, and Kalman filter tracking (below), using the graphical user interface options. Examples of different model structures are provided in the software release and below.

## Individual tracking

The classification and tracking code uses a Kalman filter [27] to keep track of individuals, assigning each a speed and heading that can be tracked even when paths overlap. Instances of multiple-detections are dealt with by combining boxes based on centroid distances and degree of overlap, and the user can select the prioritization method when assigning a class to combined detections (e.g., static dominant, motion dominant, or highest confidence), although both streams are saved in the output file.

## Annotation

The framework's annotation tool provides a user-friendly interface with initial manual annotation and training followed by optional semi-supervised annotation ("auto-annotation," Fig 7). Semi-supervised annotation allows the user to learn where the CNNs are making errors, correct them for re-training, and repeat the annotation-training cycle until the desired performance is achieved. Initial manual annotation involves selecting input videos and drawing boxes around examples of the object classes, as is standard within deep learning pipelines. The interface additionally allows the user to flip between "static" and "motion" streams, with buttons or hotkeys for assigning primary and secondary classes. Following an initial limited annotation (e.g., 50–100 annotations per class depending on variance), we recommend training the model(s) to assist with further labeling; i.e., simply run the classifier script and it will handle model training automatically (see below). The annotation script then uses the trained model(s) to predict annotations, allowing the human annotator to accept model recommendations if they are accurate, but critically, can also note any false positives or false negatives that the models make and correct them. Running the training again will now re-train the initial model. This workflow should increase the quality of the annotation dataset and speed of annotation, focusing on borderline cases and error correction, and reducing the risk of over-fitting or annotating more than is required. We also include a function for inspecting the annotated dataset, allowing for modification and error fixing.

We have introduced a feature that optionally allows the user to shield parts of frames from the training and validation data-sets by drawing a gray box over regions. Compared to a set of independent images, the frames of a video will often have

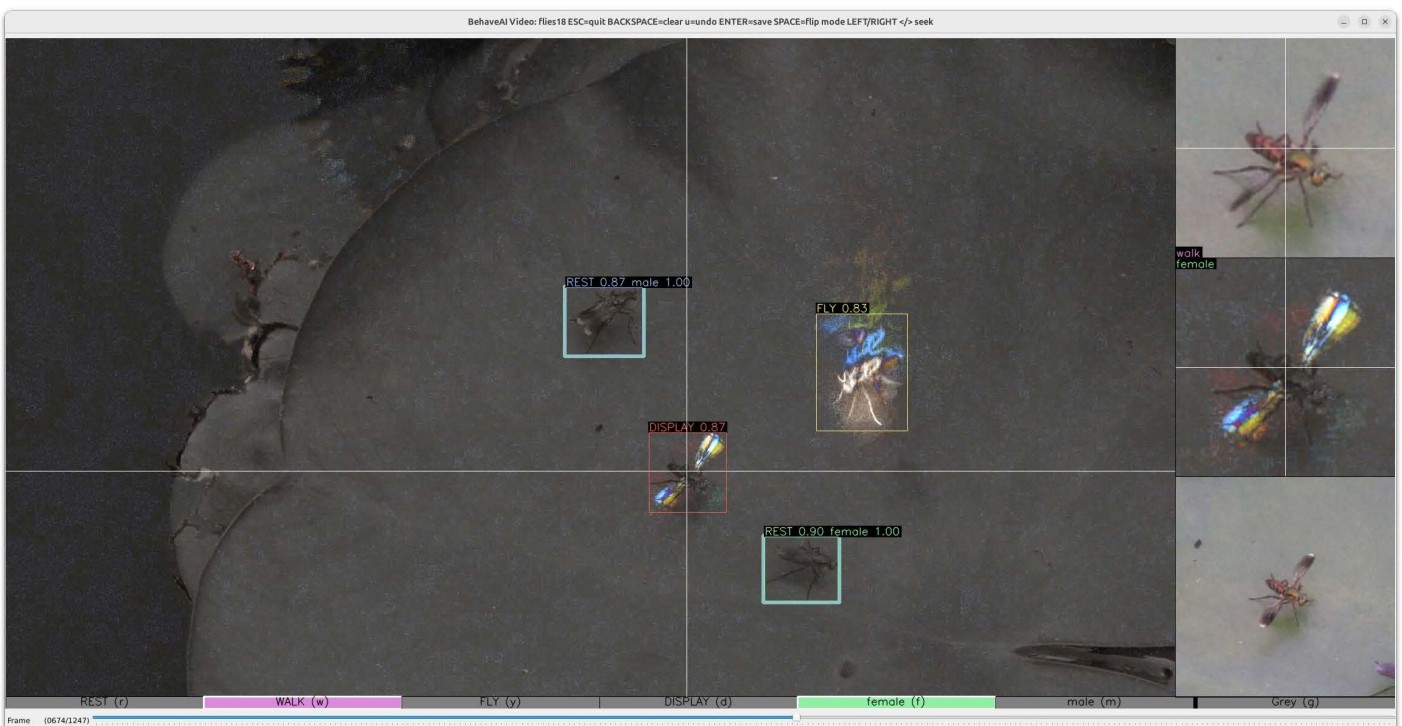

**Fig 7. Annotation user interface screenshot.** The main window shows the current motion frame (and can switch to static). The right-hand bar shows a zoom view of the current cursor position in static (top right) and motion streams (middle right), and also shows a looping animation of the video covering the same time-frame as the user-selected motion settings (bottom right). The bar at the bottom of the screen highlights the available primary (upper case) and secondary (lower case) classes, together with their associated keys. The track-bar at the bottom of the screen allows seeking through the video. Boxes in this example have all been drawn with auto-annotation, and show confidence levels for each (primary for behavior and secondary for sex).

near-identical objects and surrounds (e.g., in scenes where one target animal remains stationary while others are moving). Given all instances of a class present in any frame must be labeled for annotation, this would often require repeatedly labeling near-identical objects in successive frames. As such, models could easily become over-fitted for a specific instance and lack the diversity of training and validation data to make robust inferences in other contexts. In these cases, drawing gray boxes over the target will shield them from the model and prevent over-fitting. Drawing gray boxes is also faster than precise bounding boxes, saving annotation time. There are also instances where the human annotator cannot be sure of the class (e.g., due to partial occlusion of an animal), in which case it would be potentially counterproductive to either label the object (and run the risk of the class being incorrect and limit model precision), or leave it unlabeled so that the model learns to avoid such instances (which could also be detrimental to model recall). Using a gray box can limit the dataset to only higher-confidence instances. Importantly, gray boxes should not be used for borderline cases where one animal behavior or class is transitioning to another; it is important that the annotator uses a clear definition of the point at which these transitions occur and labels accordingly. Finally, gray boxes can also be used to shield moving instances from static frames, and/or static instances from motion frames. See the semaphore fly case study for an example where this is valuable.

### Batch classification and tracking

The classification and tracking tool manages both model training and batch input video processing, training initial models if none have been made. The tool also tracks the size of the annotation datasets and asks the user whether to re-train if additional annotations have been added. Any videos in the "input" directory are batch processed, and the tool outputs labeled videos for inspection, together with a text (".csv.") output listing each primary and (where relevant) secondary class for each individual, their centroid coordinates, plus confidence levels for each classifier.

### Assessing model efficacy

We have not undertaken direct benchmarking between the performance of BehaveAI and other techniques due to the risk of over-generalizing model utility (see Raji and colleagues [19]). Such comparisons would also raise a number of practical limitations: i) existing annotation sets would all require re-annotation for the motion stream, but ii) we aim to show that comparable performance can be achieved with much lower user annotation effort; and iii) we note that many existing datasets do not provide videos that match typical natural scenarios (complex, moving backgrounds, variable lighting), and often select behaviors based on convenience and visibility rather than end-user requirements. Instead, we report each model's precision and recall metrics.

### Case study processing

Processing was performed on a Framework 13 Laptop with Ubuntu 24.04, AMD Ryzen 7 7840U CPU (central processing unit), with NVIDIA GeForce RTX 2080 Ti, 11004MiB eGPU (external graphics processing unit, with CUDA support) unless otherwise specified.

### Raspberry Pi integration

We developed an installer and wrapper that allows for straightforward installation of the BehaveAI framework on linux systems, including Raspberry Pi OS (the tested version of Raspberry Pi OS was based on Debian 13, running on a Raspberry Pi 5 board). Raspberry Pi-specific scripts also automaticaly handle the installation picamera2.

### Supporting information

**S1 Fig. Confusion matrix and F1-confidence curves for grass moths classified from motion frames.** The data underlying this Figure are available here: https://doi.org/10.6084/m9.figshare.30531116.
(TIFF)

**S2 Fig. Confusion matrix and F1-confidence curves for grass moths classified from static frames.** The data underlying this Figure are available here: https://doi.org/10.6084/m9.figshare.30531116.
(TIFF)

**S3 Fig. Confusion matrix and F1-confidence curves for sperm classified from motion frames.** The data underlying this Figure are available here: https://doi.org/10.6084/m9.figshare.30531116.
(TIFF)

**S4 Fig. Confusion matrix and F1-confidence curves for sperm classified from static frames.** The data underlying this Figure are available here: https://doi.org/10.6084/m9.figshare.30531116.
(TIFF)

**S5 Fig. Confusion matrix and F1-confidence curves for sea slaters classified from static frames.** The data underlying this Figure are available here: https://doi.org/10.6084/m9.figshare.30531116.
(TIFF)

**S6 Fig. Confusion matrix and F1-confidence curves for sea slaters classified from motion frames.** The data underlying this Figure are available here: https://doi.org/10.6084/m9.figshare.30531116.
(TIFF)

**S7 Fig. Confusion matrix and F1-confidence curves for sea slaters classified from static frames, with training and validation datasets taken from mutually exclusive videos.** The data underlying this Figure are available here: https://doi.org/10.6084/m9.figshare.30531116.
(TIFF)

**S8 Fig. Confusion matrix and F1-confidence curves for sea slaters classified from motion frames, with training and validation datasets taken from mutually exclusive videos.** The data underlying this Figure are available here: https://doi.org/10.6084/m9.figshare.30531116.
(TIFF)

**S9 Fig. Confusion matrices and (for primary models) F1-confidence curves for semaphore flies. ( a)** primary motion model, **(b)** primary static model, **(c)** secondary model determining sex while flies are at rest, and d) while flies are walking. The data underlying this Figure are available here: https://doi.org/10.6084/m9.figshare.30531116.
(TIFF)

## Author contributions

**Conceptualization:** Jolyon Troscianko, Thomas A. O'Shea-Wheller.

**Data curation:** Jolyon Troscianko, James A. M. Galloway.

**Formal analysis:** Jolyon Troscianko, James A. M. Galloway.

**Funding acquisition:** Jolyon Troscianko, Kevin J. Gaston.

**Investigation:** Jolyon Troscianko, Thomas A. O'Shea-Wheller.

**Methodology:** Jolyon Troscianko, Thomas A. O'Shea-Wheller.

**Project administration:** Jolyon Troscianko, Kevin J. Gaston.

**Resources:** Jolyon Troscianko, James A. M. Galloway.

**Software:** Jolyon Troscianko, James A. M. Galloway.

**Supervision:** Jolyon Troscianko, Kevin J. Gaston.

**Validation:** Jolyon Troscianko, Thomas A. O'Shea-Wheller, James A. M. Galloway.

**Visualization:** Jolyon Troscianko.

**Writing – original draft:** Jolyon Troscianko, Thomas A. O'Shea-Wheller.

**Writing – review & editing:** Jolyon Troscianko, Thomas A. O'Shea-Wheller, James A. M. Galloway, Kevin J. Gaston.

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
