## [Editor Report · Decision Letter 0]

12 Nov 2025

Dear Jolyon,

Many thanks for submitting your manuscript entitled "BehaveAI: a framework for rapidly detecting and classifying objects and behaviour from motion" for consideration as a Methods and Resources paper by PLOS Biology.

Your manuscript has now been evaluated by the PLOS Biology editorial staff, as well as by an academic editor with relevant expertise, and I'm writing to let you know that we would like to send your submission out for external peer review.

Once your full submission is complete, your paper will undergo a series of checks in preparation for peer review. After your manuscript has passed the checks it will be sent out for review. To provide the metadata for your submission, please Login to Editorial Manager (https://www.editorialmanager.com/pbiology) within two working days, i.e. by Nov 14 2025 11:59PM.

Kind regards,

Roli

Roland Roberts, PhD

Senior Editor

PLOS Biology

rroberts@plos.org

---

## [Decision Letter · Decision Letter 1]

19 Dec 2025

Dear Jolyon,

Thank you for your patience while your manuscript "BehaveAI: a framework for rapidly detecting and classifying objects and behaviour from motion" was peer-reviewed at PLOS Biology. It has now been evaluated by the PLOS Biology editors, an Academic Editor with relevant expertise, and by two independent reviewers.

Based on the reviews, which you will see are highly favourable, we are likely to accept this manuscript for publication, provided you satisfactorily address the remaining points raised by the reviewers and the following data and other policy-related requests.

IMPORTANT - please attend to the following:

a) We try to avoid punctuation in Titles. Please could you change your Title to avoid this? (for example, you could simply start it with "BehaveAI is a framework for...")

b) Please address the comments from the reviewers.

c) Please address my Data Policy requests below; specifically, we need you to supply the numerical values underlying Figs 3B, S1-S9, either as a supplementary data file or as a permanent DOI’d deposition.

d) Please cite the location of the data clearly in all relevant main and supplementary Figure legends, e.g. “The data underlying this Figure can be found in S1 Data” or “The data underlying this Figure can be found in https://zenodo.org/records/XXXXXXXX

e) I note that your code is in GitHub (https://github.com/troscianko/BehaveAI); many thanks for providing this. However, because Github depositions can be readily changed or deleted, please also make a permanent DOI’d copy (e.g. in Zenodo) and provide this URL.

We expect to receive your revised manuscript within four weeks.

*Published Peer Review History*

*Press*

Sincerely,

Roli

Roland Roberts, PhD

Senior Editor

rroberts@plos.org

PLOS Biology

DATA POLICY:

Regardless of the method selected, please ensure that you provide the individual numerical values that underlie the summary data displayed in the following figure panels as they are essential for readers to assess your analysis and to reproduce it: Figs 3B, S1-S9. NOTE: the numerical data provided should include all replicates AND the way in which the plotted mean and errors were derived (it should not present only the mean/average values).

CODE POLICY

DATA NOT SHOWN?

REVIEWERS' COMMENTS:

Reviewer #1:

[identifies himself as Ammon Perkes]

Reviewing this manuscript was delightful. I want to say, up front, I think this is really excellent work. I have several suggestions below that I hope you will consider and apply, but at the risk of being called a soft reviewer, I think this already merits publication in PLOS Bio as is. I have used many different tools for tracking animal behavior and written plenty of code to do it myself, and this is by far the most effective and useful tool I have ever worked with. It reminded me a lot of the original idTracker paper, where they managed to do something seemingly miraculous just with clever feature selection. Anyway, that is probably enough effusive praise for now, but I am very excited about this tool and am very appreciative of all of the work and ingenuity that went into making it.

I've divided my review into comments about the manuscript and comments about the package itself, the latter of which is maybe slightly outside the scope of the review, but I think it could be helpful.

Concerning the manuscript:

I found the manuscript very easy to read and extremely clear. It directs people up front to the code and documentation, which is very helpful. I only have a few minor comments.

- I think the static vs motion comparisons are useful, and I agree that you don't need to do detailed performance comparisons with existing models on all these datasets. That said, I think it would be beneficial, where possible, to mention as context what the state of the field is for performance. For example, with the POP-3D dataset, you mention that your results are comparable to Chan et al, but unless I missed it, I don't think you ever state what that performance is. I think this is useful information for readers to know, with the existing caveat that this of course isn't necessarily a 1:1 comparison.

- Line 209 is missing a period and there's a "LINK" placeholder on line 248. Obviously very minor stuff, but make sure to get a careful readthrough during copy editing.

- As written, line 223 feels a bit perfunctory. I don't think you even necessarily even need to include it, but if you do, I think you could expand it slightly. I think your colour-from-motion approach is more innovative and potentially impactful than you give yourself credit, and I think it could turn out to be really powerful for a range of computer vision tasks, particularly in low-powered robotics applications.

Concerning the method, does it work as promised?

In short, yes, absolutely. I was skeptical when reading the abstract that you could really deliver what you were promising, but just in the week of playing around with it, I've been shocked at how effective it is. With around 25 frames, I was able to track a difficult dataset (baby fish swimming over gravel), where before I had used this complex combination of SLEAP trained on several hundred frames and MOG-Backround subtraction. And I was able to train and run it on my personal laptop, all in less than an hour, at 20fps. On GPUs, the speed gains are obviously dramatic, and there are two huge pipelines that I'm already in the process of moving over from SLEAP to BehaveAI because of the improved speed and accuracy that we're getting. I haven't tested the secondary classifiers as much, but already, I think this has the potential to become a mainstay of the biology world, and probably other contexts in fields I haven't even considered. It's obvious that (compared to a lot of existing packages that were designed with neuroscience or collective behavior in mind) this was really built by and for people interesting in quantifying animal behavior in a naturalistic context.

Is it sufficiently easy to use to be useful?

I found it quite easy to install based on the documentation, and a CS undergrad I asked to try it did as well. Already, this is a lot easier to use than most of the packages out there. I do have a few suggestions, but these are more like feature requests than reviewer comments, so I don't think fixing this needs to happen prior to publication. That said, I think some improvements could make this a lot more usable to the bulk of biologists, who might be less comfortable with working in the command line.

- I think the idea that each project essentially having a separate instantiation for the code is a little counter-intuitive for our app-centric world. I would suggest making this mindset slightly more clear in the documentation from the outset. My undergrad ran into trouble with some mismatches between some initial annotations and a later download of your dataset, causing batch processing to fail, which I think probably stems from not making a fresh directory.

- I think editing settings files is a pretty big barrier to entry for a lot of people, and I would really encourage you make it possible to do this from the GUI. This would also help prevent bugs caused from incorrect editing of the settings file. SLEAP for example does this really well, and is a big part of why that platform is so useful.

- I think having a written guide walking users through an example dataset with step-by-step instructions would be really helpful. The documentation almost has this already, and the video guide does this really well, but I think an explicit "label these 20 frames, now press escape, now click train & batch classify" would help get users going quickly.

- A very helpful feature would be a way to see, in the annotator, which frames have already been labeled.

- On my laptop (a macbook in darkmode) the terminal readout in the GUI is darkgrey-on-black and basically invisible. It looked great on my ubuntu desktop.

- When doing the second round of annotations, a visualization showing which frames had annotations would make it easier to see how well it worked and also help locate frames where 0 animals were detected.

- I think it could be more clear in the documentation that you need to set all the parameters in the settings file prior to running the code and starting annotations. You mention that towards the end of the documentation, but it would be helpful for that to be made clear from the start.

- The way the zoomed-in box in the annotation GUI distorts around the edges is a bit strange. I assume this is to prevent labeling outside the frame, but I wonder if there is another approach that would not distort the image.

- There are a few typos in the documentation ("determin" and "insepct", probably others)

I'm sure there are unlimited improvements you could make to this, many of which you have already considered. You already mention ideas you have for active development, and I would strongly encourage you, if you haven't already, to plan for dedicated developers whose primarily role is to improve this package and handle bugs and help requests as they arise, rather than it all falling to you to find the time in between all your other responsibilities. There are so many packages that were published but aren't actively maintained that quickly become unusable. I really do think this has the potential to be the most used tracking platform for animal behavior, but only if you can address people's challenges as they come up and continue to make it more useful over time.

Overall, I think you have created exactly what it says in the title, and I think this will be a game changer for everyone studying animal behavior, making a great deal of tracking and behavioral classification that was previously difficult or impossible fast and straightforward. I look forward to seeing all the cool new science that takes advantage of this platform.

Reviewer #2:

This is a cool paper. Its a bit out of my area of expertise, but I appreciate that the color-for-motion idea is very clever and looks like a major advance for using AI to study animal behavior from videos. This is an important topic of broad interest and looks like a good advance. Sorry I'm not qualified to comment on the technical aspects.

Introduction had a nice first few paragraphs that do a good job at reviewing the field.

The fourth intro paragraph needs some work, isn't well connected with the earlier mateiral, work on transition sentences and conceptual flow especially at the start.

The final intro paragraph is also a little confusing, as it starts by stating objectives but then starts talking about benchmarking and kind of trails off.

Figure 2 - what do the colors of the bottom frames indicate?

---

## [Editor Report · Decision Letter 2]

16 Jan 2026

Dear Jolyon,

Thank you for the submission of your revised Methods and Resources "BehaveAI enables rapid detection and classification of objects and behaviour from motion" for publication in PLOS Biology. On behalf of my colleagues and the Academic Editor, Gail Patricelli, I'm pleased to say that we can in principle accept your manuscript for publication, provided you address any remaining formatting and reporting issues. These will be detailed in an email you should receive within 2-3 business days from our colleagues in the journal operations team; no action is required from you until then. Please note that we will not be able to formally accept your manuscript and schedule it for publication until you have completed any requested changes.

Sincerely,

Roli

Senior Editor

PLOS Biology

rroberts@plos.org